# SpeS: A Novel Superantigen and Its Potential as a Vaccine Adjuvant against Strangles

**DOI:** 10.3390/ijms21124467

**Published:** 2020-06-23

**Authors:** C. Coral Dominguez-Medina, Nicola L. Rash, Sylvain Robillard, Carl Robinson, Androulla Efstratiou, Karen Broughton, Julian Parkhill, Matthew T.G. Holden, Maria R. Lopez-Alvarez, Romain Paillot, Andrew S. Waller

**Affiliations:** 1Animal Health Trust, Lanwades Park, Kentford, Newmarket CB8 7UU, UK; coral.dominguez@aht.org.uk (C.C.D.-M.); nicola.rash@aht.org.uk (N.L.R.); Robillard.sylvain@orange.fr (S.R.); carl.robinson@aht.org.uk (C.R.); maria.lopez-alvarez@aht.org.uk (M.R.L.-A.); romain.paillot@laboratoire-labeo.fr (R.P.); 2Public Health England, 61 Colindale Avenue, London NW9 5EQ, UK; androulla.efstratiou@phe.gov.uk (A.E.); karen.broughton@phe.gov.uk (K.B.); 3Wellcome Trust Sanger Institute, Cambridge CB10 1SA, UK; jp369@cam.ac.uk (J.P.); mtgh@st-andrews.ac.uk (M.T.G.H.); 4Department of Veterinary Medicine, University of Cambridge Veterinary School, Cambridge CB3 0ES, UK; 5School of Medicine, University of St Andrews, St Andrews KY16 9TF, UK; 6LABÉO Frank Duncombe, 1 Route de Rosel, CEDEX 4, 14053 Caen, France; 7Biotargen, Normandie Univ, UniCaen, 3 rue Nelson Mandela, 14280 Saint-Contest, France

**Keywords:** *Streptococcus equi*, *Streptococcus zooepidemicus*, superantigen, vaccine, adjuvant, immune response, strangles, horse

## Abstract

Bacterial superantigens (sAgs) are powerful activators of the immune response that trigger unspecific T cell responses accompanied by the release of proinflammatory cytokines. *Streptococcus equi* (*S. equi*) and *Streptococcus zooepidemicus* (*S. zooepidemicus*) produce sAgs that play an important role in their ability to cause disease. Strangles, caused by *S. equi*, is one of the most common infectious diseases of horses worldwide. Here, we report the identification of a new sAg of *S. zooepidemicus*, SpeS, and show that mutation of the putative T cell receptor (TCR)-binding motif (YAY to IAY) abrogated TCR-binding, whilst maintaining interaction with major histocompatibility complex (MHC) class II molecules. The fusion of SpeS and SpeS^Y39I^ to six *S. equi* surface proteins using two different peptide linkers was conducted to determine if MHC class II-binding properties were maintained. Proliferation assays, qPCR and flow cytometry analysis showed that SpeS^Y39I^ and its fusion proteins induced less mitogenic activity and interferon gamma expression when compared to SpeS, whilst retaining Antigen-Presenting Cell (APC)-binding properties. Our data suggest that SpeS^Y39I^-surface protein fusions could be used to direct vaccine antigens towards antigen-presenting cells in vivo with the potential to enhance antigen presentation and improve immune responses.

## 1. Introduction

*Streptococcus equi* subspecies *equi* (*S. equi*) is a host-restricted pathogen that has evolved from the zoonotic pathogen *Streptococcus equi* subspecies *zooepidemicus* (*S. zooepidemicus*) [1,2]. *S. equi* is the causative agent of strangles, a highly contagious disease characterized by pyrexia and abscessation of the lymph nodes of the head and neck in horses [3,4]. Interestingly, while these two pathogens share ≈ 80 % genomic DNA identity with the human pathogen *Streptococcus pyogenes* (*S. pyogenes*), several studies indicate that a series of genetic changes and laterally acquired elements may have influenced the evolution and pathogenic specialization of *S. equi* [5,6].

Superantigens (sAgs) are bacterial mitogens that interfere with the immune response by forming a linkage between major histocompatibility complex (MHC) class II molecules on antigen-presenting cells (APCs) and the variable region of the T cell receptor (TCR) β-chain [7]. This interaction results in a nonspecific T cell activation and overzealous release of proinflammatory cytokines such as IFN gamma (IFNγ) and tumor-necrosis factor alpha (TNFα) [8,9,10]. During a conventional immune response, only one in 10^6^ T cells is activated, whereas sAgs are capable of stimulating 20% of all the T cells, at a concentration of only 10^−15^ moles/L in culture [11]. Streptococcal toxic shock syndrome (STSS), caused by *S. pyogenes*, is the most severe presentation of invasive streptococcal infections. STSS has been associated with the production of sAgs and there is evidence that these toxins also contribute to the pathogenicity of *S. equi* and *S. zooepidemicus* [5,12,13,14,15,16].

Commons et al., 2014 proposed a single nomenclature for streptococcal sAgs [13], which we have used in this article with previous names listed in brackets. *S. equi* produces four sAgs: SpeH (SeeH), SpeI (SeeI), SpeK (SeeL) and SpeL (SeeM). These share more than 96% amino acid identity with those sAgs produced by *S. pyogenes* [13,17,18]. *S. zooepidemicus* does not cause strangles, but is the most frequently isolated opportunistic pathogen of horses and is also able to produce sAgs such as SpeN (SzeN), SpeP (SzeP), SpeO (SzeF) and orthologues of SpeK (SzeL) and SpeL (SzeM) [16,17]. The production of these sAgs, with the exceptions of SpeH and SpeO, induce the proliferation of equine peripheral blood mononuclear cells (PBMCs) [15,16]. However, some strains of *S. zooepidemicus* lack the presence of sAg-encoding genes, but possess mitogenic activity in vitro, suggesting that they produce novel sAg-encoding genes that have not yet been identified [5,16]. In particular, we previously noted that the culture media of *S. zooepidemicus* strain 1ID00103 (*Sz*1ID00103), which was isolated from the blood of a human with meningitis in 2001, had potent mitogenic activity, whilst only a copy of *speO* was identified by qPCR [16]. SpeO has little or no mitogenic activity towards equine PBMCs [16] and so we hypothesized that *Sz*1ID00103 produced an additional, as yet unidentified sAg.

Herein, we sequenced the genome of *Sz*1ID00103, identified the presence of a gene encoding a new sAg, SpeS, and report its immunological activity. Furthermore, we provide a novel method for fusing recombinant proteins from *S. equi* to a TCR-binding defective version of this sAg that may have the potential to improve the effectiveness of vaccines against strangles, which could also be applied to other infectious diseases in different animal species, including humans.

## 2. Results

### 2.1. Identification of SpeS in Streptococcus equi Subspecies Zooepidemicus Strain 1ID00103

The genome sequence of *Sz*1ID00103 was interrogated to identify novel superantigen-encoding sequences. As previously reported by Paillot et al., 2010, *Sz*1ID00103 encoded a copy of *speO* (*szeF*) [16]. However, we also identified a novel sequence encoding the putative superantigen, SpeS, which shared greatest predicted amino acid identity (66%) with SpeC of *S. pyogenes* (Figure 1A and Appendix A). SpeS contained both of the superantigen Prosite domains PS00277 and PS00278 in addition to a HxD zinc binding domain and the YAY sequence, which has been shown to be required for binding of SpeC to the TCR β-chain (Figure 1B) [19,20].

### 2.2. Disruption of the TCR-Binding Site in SpeS Impairs T Cell Proliferation In Vitro While Retaining Binding to MHC Class II Molecules

Superantigens are powerful T cell mitogens [23]. In order to determine the mitogenic activity of SpeS, equine PBMCs were analyzed after being cultured for four days with SpeS^WT^ or SpeS^Y39I^ at different concentrations. SpeS^Y39I^ contained a tyrosine to isoleucine mutation at codon 39 that changed the putative TCR-binding motif YAY to IAY. Clusters of cells were observed after 24 h in PBMCs that were cultivated with SpeS^WT^, while SpeS^Y39I^ did not stimulate T cell proliferation (Figure 2A). Proliferation induced by SpeS^WT^ was dose-dependent, with activity present at very low concentrations (1.25 × 10^−5^ μg/mL). SpeS^Y39I^ did not stimulate proliferation of PBMCs at any of the concentrations tested up to 1.25 μg/mL (Figure 2B).

We next investigated whether the ability of SpeS to bind MHC class II molecules was maintained in vitro after disruption of its predicted TCR-binding site. SpeS^WT^ and SpeS^Y39I^ were labeled with fluorescein isothiocyanate (FITC) and used to measure the level of fluorescence following binding to equine T8888 cells (MHC class II^+^) and fibroblasts (MHC class II^-^). Fluorescence was detected in T8888 cells incubated with either SpeS^WT^-FITC or SpeS^Y39I^-FITC (Figure 2C). In contrast, binding of SpeS^WT^-FITC or SpeS^Y39I^-FITC to fibroblasts lacking MHC class II was not detected (Figure 2D). Therefore, the capacity to bind to MHC class II molecules was unaffected by the Y^39^I mutation.

### 2.3. S. equi Surface Proteins Fused to SpeS^Y39I^ Are Devoid of Mitogenic Activity In Vitro

To identify if the fusion of *S. equi* surface proteins to SpeS^WT^ or SpeS^Y39I^ would affect their mitogenic activity, equine PBMCs were stimulated and analyzed after four days of culture. Six surface proteins from *S. equi* (SEQ_0402, SEQ_1959, SEQ_0933, SEQ_2190, SeM and SEQ_0232) were tested. Similar to SpeS^WT^ alone, stimulation of PBMCs with the recombinant surface proteins fused to SpeS^WT^ resulted in the formation of big clusters of cells after 24 h (Figure 3A) and stimulation index (SI) values >2. In contrast, when PBMCs were cultivated with the recombinant proteins fused to SpeS^Y39I^, the SI was reduced (Figure 3B). The type of peptide linker used (tetra-alanine or glycine-proline-glycine-proline (GPGP)) did not affect the ability of the fusion proteins to stimulate PBMCs.

### 2.4. S. equi proteins Fused to SpeS Retain Their Ability To Bind to MHC Class II Molecules

Based on the T cell proliferation results, we reduced our candidates to four using only those fused by the tetra-alanine (ala) peptide linker and measured whether fusion of these surface proteins to SpeS could enhance binding to MHC class II^+^ molecules compared to the sAgs alone. To establish the level of binding, the fusion protein was labeled with FITC and fluorescence was assessed by flow cytometry in MHC class II^+^ and MHC class II^-^ equine cells. While MHC class II^+^ cells were targeted by the fusion *S. equi* proteins-FITC, the level of fluorescence (Figure 4A) was slightly weaker than that observed with the SpeS^WT^-FITC or SpeS^Y39I^-FITC alone (Figure 2C). No binding by the fusion proteins was detected when MHC class II^-^ cells were used (Figure 4B).

### 2.5. Expression of IFNγ Decreased In Vitro After Stimulation with S. equi Surface Proteins Fused SpeS^Y39I^

In order to determine if the lack of mitogenicity correlated with cytokine levels, we determined the effect of stimulation with the fusion proteins on the expression of cytokines by PBMCs. The selection of targets primarily included some of the cytokines that act as mediators of toxic shock syndrome [24,25,26,27]. The effect of the proteins fused to SpeS^WT^ in PBMCs was compared to responses following exposure to the proteins fused to SpeS^Y39I^. After 16 hours, we clearly observed higher levels of IFNγ mRNA in cells that were stimulated with the surface proteins fused to SpeS^WT^ relative to those fused to SpeS^Y39I^; however this was not statistically significant (Figure 5A). Levels of IFNγ mRNA reached their peak at 24 h where cells exposed to SpeS^WT^ and the WT fusion proteins showed an increment of more than double of the IFNγ expression relative to the Y^39^I SpeS fusions (Figure 5B). After 48 h, cells stimulated with the WT SpeS fusions continued to show high levels of IFNγ (Figure 5C) followed by a decrease at 72 h (Figure 5D) when all the groups showed similar levels of IFNγ mRNA expression. A significant reduction in the mRNA expression of TNFα and IL-6 was observed in PBMCs stimulated with most of the SpeS^WT^ fusion proteins after 48 or 72 h, respectively (Appendix A). The expression of IL-10 did not vary significantly between groups at any time point, except at 24 h for the SEQ_0933-SpeS fusion proteins (Appendix A).

## 3. Discussion

sAgs play an important role in the disease of animals and humans by interfering with the development of protective immune responses through the generation of uncontrollable proinflammatory mitogenic responses [13]. In the case of *S. equi*, the sAgs SpeI, SpeH, SpeK and SpeL are associated with its ability to cause strangles and lymph node abscesses, but its mitogenicity is dependent on the activity of all except SpeH [15]. Interestingly, the presence of sAgs in *S. zooepidemicus* isolates has been associated with non-strangles cases of lymph node abscessation [14] and mitogenic activity has been attributed to the sAgs SpeK, SpeL, SpeN and SpeP [16,17,28]. Nevertheless, genes encoding these sAgs were not identified in the genomes of several *S. zooepidemicus* strains that had mitogenic activity [5,16]. From the 165 *S. zooepidemicus* isolates sequenced by Paillot et al., 2010 [16], 12 contained *speO* alone and were mitogenic. SpeO was reported to lack mitogenic activity [14,16] suggesting the presence of new genes. From these isolates we sequenced the strain *Sz*1ID00103, which significantly stimulated equine PBMCs in vitro, and identified a new potent mitogenic sAg, which we have named SpeS.

When compared to other sAgs, we found that SpeS shared 66% amino acid identity with SpeC of *S. pyogenes* and that the T cell receptor ß-binding motif in SpeS was homologous to that of SpeC (YAY) (Figure 1). Yamaoka et al., 1998 showed that SpeC mutants where the tyrosine (Y), alanine (A) and tyrosine (Y) were replaced with isoleucine (I), glutamic acid (E) and isoleucine (I) in positions 15, 16 and 17, respectively, had significantly reduced mitogenic activity [20]. For our study, we generated a Y^39^I mutant of SpeS (YAY to IAY) which completely impaired T cell proliferation in PBMCs at all concentrations. Similar to the results showed by Yamaoka et al., 1998, while mutation of the TCR-binding site in SpeS reduced T cell proliferation, its ability to bind to MHC class II molecules remained intact.

The application of modified sAgs in therapeutic interventions to stimulate protective antibody responses has been determined [29,30,31]. A study by Zeppa et al., 2017 used a nonfunctional version of the sAg SpeA of *S. pyogenes* to confer protection to mice after nasopharyngeal infection [29]. Furthermore, due to their effects on APCs, the potential of using sAgs as vaccine adjuvants by coupling TCR-binding defective sAgs to other molecules to directly deliver antigens to APCs by targeting elements on the cell surface opened the possibility of using SpeS^Y39I^ as a targeting agent against strangles. For instance, mice vaccinated with the TCR-binding deficient sAg, SMEZ-2 conjugated to ovalbumin showed a 1000- to 10,000-fold increase in IgG titers with a significant improvement in antigen presentation and T cell responses with an enhanced antitumor activity [32,33]. Moreover, the potential of cross-presentation by SMEZ-2 conjugates was demonstrated by pulsing dendritic cells with SMEZ-2 coupled to the nucleocapsid of hepatitis B virus, which stimulated T cells more effectively than with the capsid alone [34].

Currently there is no effective vaccine against strangles that provides long-lasting immunity, but vaccination studies in horses have indicated the potential of multicomponent subunit vaccines, [35,36,37]. In our study, the aim of using surface proteins from *S. equi* fused to SpeS^Y39I^ was to deliver a prototype vaccine to target multiple immunogenic components on the cell surface to generate a better immune response. In support of this strategy, the analysis of the crystal structure of SpeC performed by Li et al., 2001 [38] suggested the existence of a similar site in the C-terminal region of SpeS available for fusing other proteins that could potentially mimic antigen presentation and stimulation of T cells in a conventional way. The sortase-processed surface proteins selected here were previously identified as immunogenic targets or important components in strangles diagnosis tests [35,36,37,39].

Additionally, the use of peptide linkers to fuse proteins has been shown to minimize changes in the protein structure and, in some cases, provide flexibility to the peptide chain [40]. Therefore, we selected the tetra-alanine (rigid) and GPGP (flexible) linkers and, while some studies support the use of rigid linkers [41,42,43] over the two linkers used here, neither made a difference in our in vitro studies. Interestingly, the fusion proteins did not alter the activity of SpeS as SpeS^WT^ fusions retained the ability to stimulate T cells and bind to MHC class II molecules, whereas those fused to SpeS^Y39I^ did not have any mitogenic activity but retained the ability to bind to MHC class II molecules.

Finally, we determined how the fusion proteins affected the cytokine response generated by SpeS. Paillot et al., 2010 showed that the T cell proliferative response induced by *S. equi* sAgs indicated that the early synthesis of IFNγ could play an important role in the pathogenesis of strangles [15] but its direct effect in such response remained to be determined. Here we report a similar response in which IFNγ levels peaked after two days, but decreased thereafter. Furthermore, the removal of the ability to bind to TCRs in SpeS was enough to suppress IFNγ production in the presence of SpeS^Y39I^ and its fusion proteins, suggesting that they could be safe for in vivo studies. However, the consistent expression of TNFα and IL-6 for up to 24 h after exposure to SpeS^Y39I^ suggests that the capacity of superantigens to elicit these cytokines is independent of their interaction with the TCR. Although, the superantigen–TCR binding function might be involved in regulatory mechanisms since the presence of the wild-type SpeS fusion proteins significantly reduced the expression of TNFα and IL-6 at later time points. A deeper study in how the kinetics of the cytokine response develops in vivo may be useful to understand better the function of SpeS.

There are studies that indicate that a difference in the cytokine secretion profiles may be dependent on the type of sAg [25,26,44] as well as their affinity to the TCR Vβ and MHC class II repertoires and associated T lymphocyte diversity [45]. This could be a reason why we could not detect high levels of any of the other cytokines studied here following stimulation of equine PBMCs with SpeS. Additionally, for *S. equi* infection it has been demonstrated that sAgs have localized active mitogenic activity that result mainly in TNFα expression in abscesses material and chondroids [12]. This suggests that systemic sAgs dissemination may be unnecessary or avoided and that the site of infection and associated immune cells and tissues could also be influencing the synthesis of different cytokines. The measurement of cytokine levels at earlier time points may be needed, including the use of different detection methods. Our results show that fusing surface proteins to SpeS may provide access to a new method of antigen delivery to APCs, however, the activity of these fusion proteins in vivo still requires further study.

## 4. Materials and Methods

### 4.1. Bacterial Strains and DNA Extraction

Strain SzID00103 was recovered from the blood of a human case of meningitis in the UK during 2001 and grown on Todd-Hewitt agar in a humidified atmosphere at 37 °C containing 5% CO_2_. A single colony was inoculated into Todd-Hewitt broth (THB) and grown overnight in at 37 °C in a 5% CO_2_-enriched atmosphere. Cells were harvested, and chromosomal DNA was extracted according to the method of Marmur [46] with the addition of 5000 units of mutanolysin (Sigma-Aldrich, Gillingham, Dorset, UK) and 20 μg of RNase A (Sigma-Aldrich) during the lysis step. DNA was initially sequenced using a Genome Sequencer-FLX (454 Life Sciences, Roche Applied Sciences, Branford, CT, USA). This strain was recently resequenced on an Illumina HiSeq DNA sequencer (Illumina, San Diego, CA, USA) and is available under the accession number ERS134301. Putative sAg-encoding genes were identified in the *Sz*1ID000103 genome by searching for coding sequences with similarity to known sAg-encoding genes.

### 4.2. Cloning and Purification of speS and Recombinant Fusion Proteins

*speS* was amplified from *Sz*1ID00103 DNA using the primers listed in Appendix A and Phusion DNA polymerase (New England Biolabs, Ipswich, MA, USA) and cloned into pGEX-3X (GE Healthcare, Little Chalfont, Buckinghamshire, UK). Replacement of a tyrosine with isoleucine at amino acid position 39 (Y^39^I) in *speS* was carried out based on its predicted amino acid sequence identity with the TCR-binding site of SpeC of *S. pyogenes* [20]. Primers encoding the desired modification (Appendix A) were used to mutate the plasmid containing the *speS* wild-type sequence. The resulting products were digested with DpnI to remove carried over parental plasmid DNA and the mutated *speS* gene was transformed into competent cells and sequenced using an ABI3100 DNA sequencer (Applied Biosciences, Waltham, MA, USA) and data were analyzed with Seqman (DNASTAR, Madison, WI, USA).

DNA encoding *S. equi* surface proteins was amplified from *S. equi* strain 4047 (*Se*4047) genomic DNA and fused to SpeS^WT^ or SpeS^Y39I^ by PCR. To enable movement between SpeS and the *S. equi* recombinant proteins, DNA encoding a tetra-alanine or GPGP peptide linker was incorporated between them. Each of the *speS* and surface protein-encoding genes were designed to produce gluthatione S-transferase (GST) fusions where the signal- (for both) and sortase-motif (for the surface proteins) sequences were omitted. The final PCR products were cut with BamHI and EcoRI (New England Biolabs), cloned into pGEX-3X (GE Healthcare) and sequenced as described above. The primers used for cloning and sequencing are listed in Appendix A.

Plasmids were transformed into *E. coli* BL21-DE3 and selected by growth at 37 °C in 2× YT containing 50 µg/mL ampicillin. The production of recombinant proteins was induced by the addition of 1 mM of IPTG and the cultures incubated at 4 h at 28 °C [15]. Cells were harvested and lysed by digestion with lysozyme for 1 h at 4 °C followed by sonication and the GST-proteins were purified over glutathione Sepharose 4B beads according to the supplier’s protocol (GE Healthcare). Factor Xa (Amersham Biosciences, Little Chalfont, Buckinghamshire, UK) was then used to cleave the recombinant proteins from the GST tag. The concentrations of the purified recombinant proteins were quantified, and the proteins were stored at −70 °C in 50% glycerol. The purity of the proteins was assessed by polyacrylamide gel electrophoresis and staining with Coomassie blue. All of the recombinant proteins migrated at the expected sizes and were shown to be greater than 99% pure.

### 4.3. Stimulation of Peripheal Blood Mononuclear Cells and Thymmidine Incorporation Assay

Equine PBMCs were isolated from heparinized blood by Ficoll density gradient centrifugation and cultured in RPMI media containing 10% heat inactivated foetal calf serum (FCS), 2 mM l-glutamine, 55 µM 2β-mercaptoethanol and 1% penicillin/streptomycin (all supplied by Sigma, Poole, UK). Medium alone was used as a negative control. For dose-dependent assays, a total of 10^6^× equine PBMCs were stimulated with 1/10 dilutions of SpeS^WT^ or SpeS^Y39I^ (1.25E-00–1.25E-07 μg/mL) for four days. Pictures of the culture plates were taken at 24, 48, 72 and 96 h. For proliferation assays using the fusion proteins a volume of 200 μl of PBMCs at 1 × 10^6^ cells/mL was cultured in triplicate with the recombinant fusion proteins, SpeS^WT^ or SpeS^Y39I^ at the indicated concentrations in flat-bottom 96-well plates.

Proliferation of PBMCs was quantified by [^3^H]thymidine (^3^HT) incorporation as previously described [15]. After 72 h, proliferation of equine PBMCs was measured by overnight incorporation of 0.5 μCi of ^3^HT (Amersham Biosciences) into new strands of chromosomal DNA during mitotic cell division. Radioactivity was measured in a scintillation ß-counter (TopCount NXT, Perkin Elmer, Chalfont Road, Buckinghamshire, UK). The stimulation index (SI) was calculated as follows: (Response of stimulated PBMCs) / (Response of unstimulated PBMCs).

### 4.4. Fluorescein Isothiocyanate (FITC) Labelled Recombinant Fusion Proteins and Detection of MHC Class II Binding

Recombinant fusion proteins and SpeS^WT^ or SpeS^Y39I^ were conjugated to FITC using a FluoReporter FITC Protein Labelling Kit (Invitrogen, Waltham, MA, USA) according to the manufacturer’s instructions. Binding to MHC class II molecules was determined by stimulating equine lymphoid cells (EqT8888 cell line) [47] and equine fibroblasts, at a concentration of 1 × 10^6^ cells/mL, with the FITC-labelled proteins for 20 min at 4 °C. EqT8888 cells, which produce MHC class II molecules, were cultured in RPMI (44%) and AIM-V (55%) media containing 10% FCS, 2 mM l-glutamine, 0.5% nonessential amino acids, 0.5% sodium pyruvate, 50 µg/mL gentamycin and 55 µM 2β-mercaptoethanol (all supplied by Sigma-Aldrich). Primary equine fibroblasts that lack the production of MHC class II molecules were purified from skin punch biopsies as previously described [48]. Fibroblasts were cultivated in complete MEM medium (cMEM) containing 10% FCS, 2 mM l-glutamine, 1 mM sodium pyruvate, 55 µM 2β-mercaptoethanol, 0.25 µg/mL amphotericin B and 1% penicillin/streptomycin (all supplied by Sigma-Aldrich). Cells were analyzed on an ACEA NovoCyte 2000R flow cytometer (ACEA Biosciences, San Diego, CA, USA).

### 4.5. Measurement of IFNγ, TNFα, IL-10 and IL-6 Expression by qPCR

A total of 4 × 10^6^ cells per well were stimulated with the recombinant fusion proteins, SpeS^WT^ or SpeS^Y39I^ (1 μg/mL), or PBS and incubated at 37 °C for 16, 24, 48 and 72 h. Cells were resuspended in lysis buffer and stored at −70 °C until needed. RNA extraction was performed with a NucleoSpin RNA Kit (Macherey-Nagel, Dueren, Germany). RNA samples were diluted to a concentration of 0.5 ng per well and DNase treated using a High Capacity cDNA Transcription Kit (Thermo Fisher Scientific, Vilnius, Lithuania) following the manufacturer’s instructions. Forward and reverse primers were designed to specifically amplify each gene (Appendix A). Cytokine expression was determined using KAPA SYBR Fast Universal Mastermix (Sigma-Aldrich) in a final volume of 10 μL per well as described previously [14]. A total of 0.5 ng/well of cDNA was amplified under the following conditions: 3 min at 95 °C for enzyme activation, 40 cycles of 3 s at 95 °C for denaturation, and 40 s at 60 °C for anneal and extension, followed by 0.5 °C increments with a SYBR read at each step using a CFX96 Touch Real Time System (Bio-Rad, Watford, Hertfordshire, UK). Gene expression was determined by the comparative threshold cycle method (Ct or ΔΔCt) where each sample was normalized against the average of two reference genes, B-actin and GAPDH, and differences between the experimental and untreated (control) samples were calculated.

### 4.6. Statistical Analysis

Data was analyzed using GraphPad Prism v. 6.0. The Mann–Whitney U test was used to determine differences in nonparametric data. All *p*-values less than 0.05 were considered significant.

## Figures and Tables

**Figure 1 ijms-21-04467-f001:**
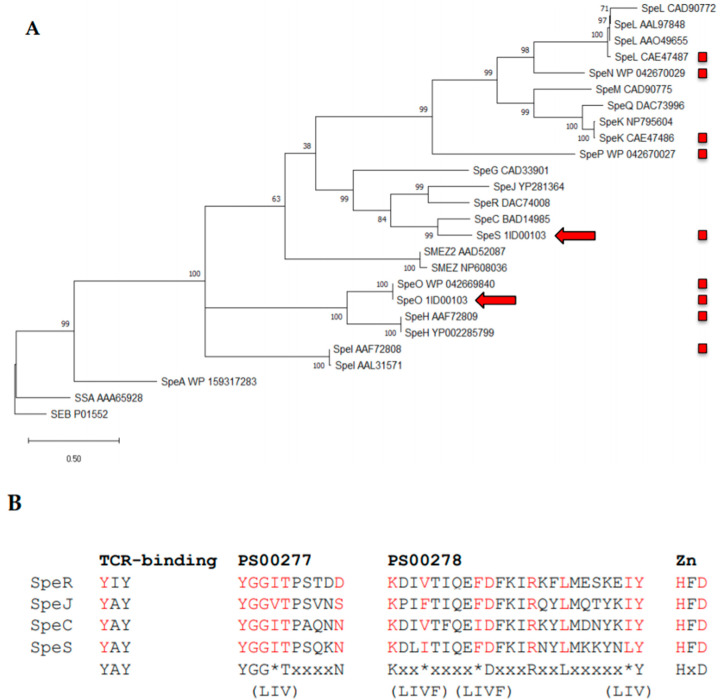
Similarity of the novel superantigen SpeS to known superantigens. (**A**) The amino acid sequences of representatives of the 16 known streptococcal superantigens, SpeS and the staphylococcal superantigen SEB were obtained. Nomenclature has been standardized in accordance with Commons et al., 2014 [13]. The signal sequences were identified using SignalP [21] and removed. The remaining predicted amino acid sequences were utilized to generate a maximum likelihood tree in MEGA X [22], which was rooted to SEB. The percentage of trees in which the associated taxa clustered together is shown next to the branches. The tree is drawn to scale, with branch lengths measured in the number of substitutions per site. The red arrows indicate the location of the SpeO and SpeS sequences encoded within the *Sz*1ID00103 genome. Red boxes indicate superantigens encoded by *S. equi* or *S. zooepidemicus*. (**B**) Identification of signature domains found in superantigens. The amino acid sequence of SpeS was compared to the closest relatives SpeC, SpeJ and SpeR to identify motifs typically found in superantigens. The T cell receptor ß-binding motif YAY of SpeC [19,20], the consensus sequence Y-G-G-(LIV)-T-x_4_-N and K-x_2_-(LIVF)-x_4_-(LIVF)-D-x_3_-R-x_2_-L-x_5_-(LIV)-Y of the superantigen Prosite domains PS00277 and PS00278, respectively, and the zinc binding domain, HxD, were fully conserved in SpeS. Consensus matching residues are indicated in red. * represents one of the three or four possible amino acids indicated in brackets on the line directly below.

**Figure 2 ijms-21-04467-f002:**
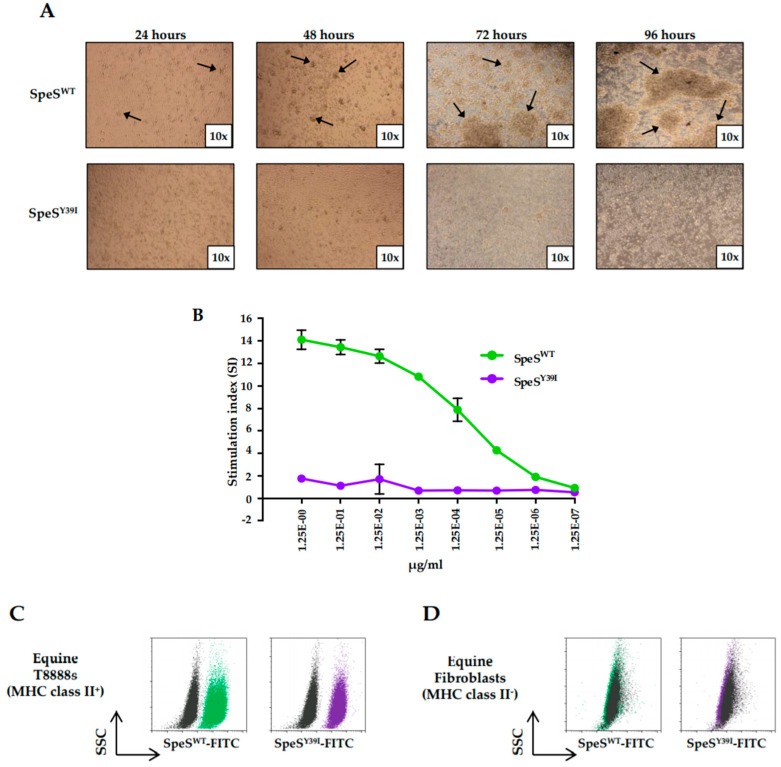
SpeS^Y39I^ does not stimulate T cell proliferation but maintains its ability to bind major histocompatibility complex (MHC) class II molecules in vitro. (**A**) Images of equine peripheral blood mononuclear cells (PBMCs) stimulated with 1 μg/mL of SpeS^WT^ or SpeS^Y39I^ for 24, 48, 72 and 96 h. Cells were captured with a camera at 10× magnification. Pointed arrows indicate the formation of clusters of cells after stimulation with SpeS^WT^. (**B**) Dose response; a total of 1 × 10^6^ cells/mL PBMCs from two different donors were cultured in triplicate with 1/10 dilutions of SpeS^WT^ or SpeS^Y39I^ for four days and incubated with ^3^HT for 16 h prior quantification of proliferating cells. Results are shown as stimulation index (SI) and error bars represent the standard error of the mean (SEM). (**C**,**D**) Flow cytometry analysis of SpeS^WT^ (green) and SpeS^Y39I^ (purple) binding to MHC class II molecules. (**C**) Equine T8888s and (**D**) fibroblasts were incubated for 20 min at 4 °C with SpeS^WT^-Fluorescein isothiocyanate (FITC) or SpeS^Y39I^-FITC in PBS, washed and the level of fluorescence was detected. Unstained cells (black) were used as a negative control.

**Figure 3 ijms-21-04467-f003:**
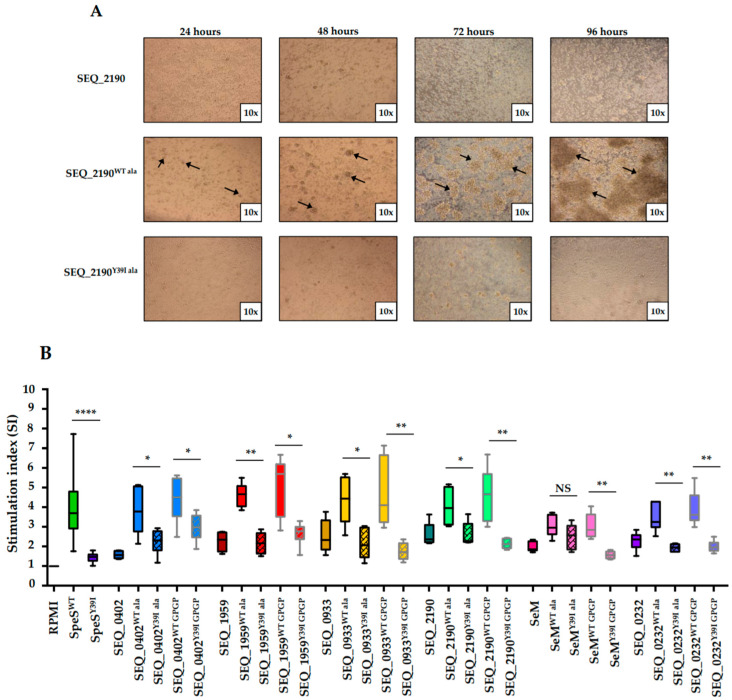
*S. equi* surface proteins fused to SpeS^Y39I^ lack mitogenic activity in vitro. (**A**) Images of equine PBMCs cultured for 24, 48, 72 and 96 h with 1 μg/mL of SEQ_2190 or SEQ_2190 fused to SpeS^WT^ or SpeS^Y39I^. Images are representative of all the *S. equi* surface proteins and fusion proteins used. Photos of the cells were taken with a camera at 10× magnification. Pointed arrows show the clusters of cells formed after being exposed to the SpeS^WT^ fusion protein. (**B**) Equine PBMCs (1 × 10^6^ cells/mL) from two different donors were stimulated in triplicate with 1 μg/mL of the *S. equi* surface proteins alone or as fusion proteins for four days. ^3^HT was added for 16 h before measuring the proliferation of cells. SpeS^WT^, SpeS^Y39I^ and medium alone were used as controls. Results are presented as SI. Data is from three independent experiments. Error bars correspond to SEM. * *p* ≤ 0.05, ** *p* ≤ 0.01, **** *p* ≤ 0.0001.

**Figure 4 ijms-21-04467-f004:**
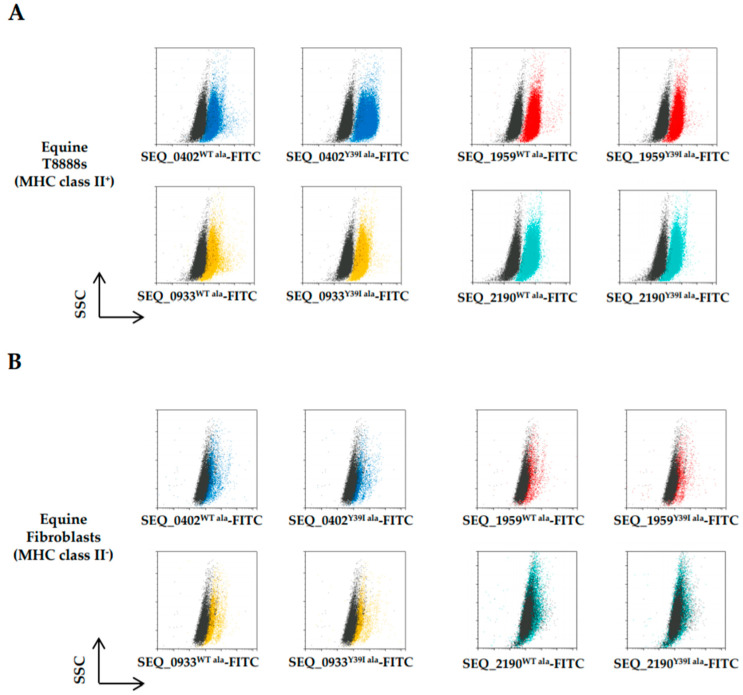
*S. equi* surface proteins fused to SpeS^WT^ or SpeS^Y39I^ retain their ability to bind to MHC class II molecules in vitro. Flow cytometry analysis of SEQ_0402 (blue), SEQ_1959 (red), SEQ_0933 (yellow) and SEQ_2190 (aqua green) fusion proteins binding to MHC class II molecules. (**A**) Equine T8888s and (**B**) fibroblasts were incubated for 20 min at 4 °C with the FITC-labelled proteins, washed and the level of fluorescence analyzed. Unstained cells (black) were used as a negative control.

**Figure 5 ijms-21-04467-f005:**
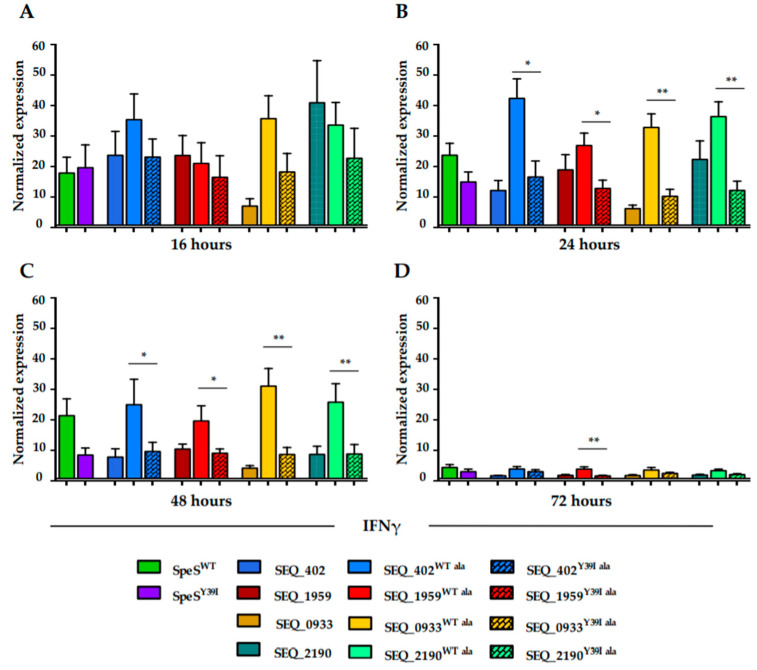
Levels of IFNγ (a proinflammatory cytokine) mRNA expression are reduced in PBMCs stimulated with SpeS^Y39I^ and *S. equi* surface proteins fused to SpeS^Y39I^. A concentration of 4 × 10^6^ cells was cultured in duplicate with 1 μg/mL of PBS (control), SpeS^WT^, SpeS^Y39I^ or the fusion proteins for (**A**) 16 h, (**B**) 24 h, (**C**) 48 h and (**D**) 72 h. Samples were normalized against the housekeeping genes β-actin and GAPDH. Differences between the experimental samples and control samples were calculated. Results are from three different experiments where PBMCs from two different donors were used. Error bars represent SEM. * *p* ≤ 0.05, ** *p* ≤ 0.01.

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
