# Peer review of "SpeS: A Novel Superantigen and Its Potential as a Vaccine Adjuvant against Strangles"

_ijms, 2020, doi:10.3390/ijms21124467_

Round 1
Reviewer 1 Report
In “SpeS: A Novel Superantigen and Its Potential as a Vaccine Adjuvant against Strangles,” Dominguez-Medina et al. sequence a new isolate of S. equi to identify a superantigen, which they propose is sufficiently different from known superantigens, and name it SpeS. In a series of molecular experiments, they show that SpeS binds MHC-II and is sufficient to induce T cell proliferation and release of IFNg, activities of true superantigens. The experiments are relatively straightforward, appropriately controlled, and the paper is generally well written. The methods are frequently inadequately detailed, and some claims may require further support. These are detailed as following points which should be readily addressable by the authors. This is on the whole a strong paper I recommend for publication with minor revision.
Major:
‘Fibroblasts’ are used as a negative control for MHC-II binding. Descriptions of these cells source and their culture and treatment is lacking. They are not confirmed to be MHC-II negative, and though it is expected of a fibroblast, the lack of provenance of these cells gives concern.
The data presented on SpeS is not sufficient to state that it is sufficiently distinct from SpeC or other superantigens that it should be considered a new one. The reported 66% amino acid identity would be suggestive of that, but this lacks context. The dendrogram is inconsistent in how many isolates of each group are included, and appears to be inconsistent between which species or clones are included. The major utility of the dendrogram is to make the argument that SpeC is the most appropriate one to compare SpeS to, but this is not clear. From there, a deeper analysis of the diversity among SpeC tells how well SpeS fits in. Furthermore, it would be useful to note whether SpeS is chimeric or whether divergent homology is localized within a particular region of the protein. For being a ’new’ superantigen, this is an inadequately described protein.
The Y39I protein, which does bind MHC-II, is inconsistent in activities - it stimulates T cell proliferation, despite not bridging an interaction with antigen-presenting cells, but still induces T cell IFNg. Do the authors have an explanation for this? There also seems to be inconsistency in stimulating proliferation, as the difference SpeS/Y39I is ~3 x in Fig 3B, but substantially greater at the same concentration under the same protocol in Fig 2B. This variation is much greater than would be expected, or is presented in the graphs.
Minor:
- No sequence provided for this ’novel’ superantigen or parent genome, nor indication of deposition in an available database.
- Line 82: The utility of the Y39I protein would be more clear if how it was designed was explained before this
- Fig 3A. Identity of the “protein" is unclear
- All microscopy lacking scale bar Line 190 onward, this description of cytokine profiles is vague and should be discussed in greater detail
- The method for normalization of the qPCR data is unclear, in particular how data are normalized to both actin and GAPDH, and the specific identity of control samples (an untreated sample, and the same for all experiments?)
- Method for calculating stimulation index should be more clear - is this based on counting cell number?
- No binding to TCR is shown. While there is a functional readout T cell activation, this would strengthen the paper in showing the a superantigen-mediated synapse is formed.
- From the protocol, it appears these proteins all still contain a GST tag and polylinker. If so, this should be noted on the proteins, as they are not equivalent to native SpeS. Discussion of tag location and whether it may interfere in any binding should also be included.
Author Response
Reviewer 1
- In “SpeS: A Novel Superantigen and Its Potential as a Vaccine Adjuvant against Strangles,” Dominguez-Medina et al. sequence a new isolate of S. equi to identify a superantigen, which they propose is sufficiently different from known superantigens, and name it SpeS. In a series of molecular experiments, they show that SpeS binds MHC-II and is sufficient to induce T cell proliferation and release of IFNg, activities of true superantigens. The experiments are relatively straightforward, appropriately controlled, and the paper is generally well written. The methods are frequently inadequately detailed, and some claims may require further support. These are detailed as following points which should be readily addressable by the authors. This is on the whole a strong paper I recommend for publication with minor revision.
> We thank this reviewer for their comments and recommendations, which we believe have improved our manuscript
Major:
- ‘Fibroblasts’ are used as a negative control for MHC-II binding. Descriptions of these cells source and their culture and treatment is lacking. They are not confirmed to be MHC-II negative, and though it is expected of a fibroblast, the lack of provenance of these cells gives concern.
> We have modified the manuscript to include a description of the origins of the fibroblast cells, which were confirmed to lack production of MHC class II by FACS (see below).
- The data presented on SpeS is not sufficient to state that it is sufficiently distinct from SpeC or other superantigens that it should be considered a new one. The reported 66% amino acid identity would be suggestive of that, but this lacks context. The dendrogram is inconsistent in how many isolates of each group are included, and appears to be inconsistent between which species or clones are included. The major utility of the dendrogram is to make the argument that SpeC is the most appropriate one to compare SpeS to, but this is not clear. From there, a deeper analysis of the diversity among SpeC tells how well SpeS fits in. Furthermore, it would be useful to note whether SpeS is chimeric or whether divergent homology is localized within a particular region of the protein. For being a ’new’ superantigen, this is an inadequately described protein.
> We have added a supplementary Figure (new Supplementary Figure 1), which shows that differences between SpeS and SpeC occur throughout the protein and are not concentrated within a particular region suggestive of a chimeric protein. The dendrogram includes examples of representative superantigens encoded by S. pyogenes, S. equi or S. zooepidemicus. For clarification, the dendrogram in Figure 1 has been updated to highlight those sAgs that were identified in S. equi or S. zooepidemicus.
- The Y39I protein, which does bind MHC-II, is inconsistent in activities - it stimulates T cell proliferation, despite not bridging an interaction with antigen-presenting cells, but still induces T cell IFNg. Do the authors have an explanation for this? There also seems to be inconsistency in stimulating proliferation, as the difference SpeS/Y39I is ~3 x in Fig 3B, but substantially greater at the same concentration under the same protocol in Fig 2B. This variation is much greater than would be expected, or is presented in the graphs.
Minor:
- No sequence provided for this ’novel’ superantigen or parent genome, nor indication of deposition in an available database.
> The genome sequence of strain Sz1ID00103 is available under the accession number ERS134301 (Line 224).
- Line 82: The utility of the Y39I protein would be more clear if how it was designed was explained before this.
> Sentences introducing the concept of TCR disruption and the YAY TCR-binding motif are present on lines 70 and 80. The actual design of the surface protein fusions is provided in the Materials and Methods section, which we hope is appropriate.
- Fig 3A. Identity of the “protein" is unclear
> Apologies, the identity of the surface protein used, SEQ2190, has been added to this Figure as suggested.
- All microscopy lacking scale bar
> We have added the magnification that was used on the microscopy images. These images are included to illustrate the proliferation of PBMCs in response to the presence of a wild-type SpeS, but not the SpeSY39I without and with fusion to SEQ2190.
- Line 190 onward, this description of cytokine profiles is vague and should be discussed in greater detail
> We have modified this section to provide further detail as suggested by this reviewer:
Finally, we determined how the fusion proteins affected the cytokine response generated by SpeS. Paillot et al., 2010 showed that the T cell proliferative response induced by S. equi sAgs indicated that the early synthesis of IFNγ could play an important role in the pathogenesis of strangles [15] but its direct effect in such response remained to be determined. Here we report a similar response in which IFNγ levels peaked after 2 days, but decreased thereafter. Furthermore, the removal of the ability to bind to TCRs in SpeS was enough to suppress IFNγ production in the presence of SpeSY39I and its fusion proteins, suggesting that they could be safe for in vivo studies. However, the consistent expression of TNFα and IL-6 for up to 24h post-exposure to SpeSY39I suggests that the capacity of superantigens to elicit these cytokines is independent of their interaction with the TCR. Although, the superantigen-TCR binding function might be involved in regulatory mechanisms since the presence of the wild-type SpeS fusion proteins significantly reduced the expression of TNFα and IL-6 at later time points. A deeper study in how the kinetics of the cytokine response develops in vivo may be useful to understand better the function of SpeS.
- The method for normalization of the qPCR data is unclear, in particular how data are normalized to both actin and GAPDH, and the specific identity of control samples (an untreated sample, and the same for all experiments?)
> Apologies, we have modified this section to clarify:
“Gene expression was determined by the comparative threshold cycle method (Ct or ΔΔCt) where each sample was normalized against the average Ct of two reference genes, B-actin and GAPDH, and differences between the experimental and untreated (control) samples were calculated.”
- Method for calculating stimulation index should be more clear - is this based on counting cell number?
> Apologies we have modified the methods section to provide clarification:
“Proliferation of PBMCs was quantified by [3H]thymidine (3HT) incorporation as previously described [15]. After 72 hours, proliferation of equine PBMCs was determined by overnight incorporation of 0.5 μCi of 3HT (Amersham Biosciences) into new strands of chromosomal DNA during mitotic cell division. Radioactivity was measured in a scintillation β-counter (TopCount NXT, Perkin-Elmer). The stimulation index (SI) was calculated as follows: (Response of stimulated PBMCs)/(Response of unstimulated PBMCs).”
- No binding to TCR is shown. While there is a functional readout T cell activation, this would strengthen the paper in showing the a superantigen-mediated synapse is formed.
- From the protocol, it appears these proteins all still contain a GST tag and polylinker. If so, this should be noted on the proteins, as they are not equivalent to native SpeS. Discussion of tag location and whether it may interfere in any binding should also be included.
> Apologies, the GST tag was cleaved from the N-terminus of all of the recombinant proteins using Factor Xa. The methods section has been modified to provide clarification.
Reviewer 2 Report
The paper reports an in-depth characterization of a new S. equi superantigen. Moreover, it describes an interesting new method for antigen delivery to APCs. The methods are well described and data supports the presented results.
Author Response
Reviewer 2
- The paper reports an in-depht characterization of a new S. equi superantigen. Moreover, it describes an interesting new method for antigen delivery to APCs. The methods are well described and data supports the presented results.
> We are grateful to this reviewer for their comments.
Reviewer 3 Report
In this manuscript, the authors report the identification of new bacterial superantigens (sAgs) of S. zooepidemicus, SpeS, and show that mutation of the putative T cell receptor (TCR)-binding motif (YAY to IAY) abrogated TCR-binding, whilst maintaining interaction with major histocompatibility complex (MHC) class II molecules. It is interesting, but the authors missed a very important functional assay for the newly identified sAgs. SpeS was identified based on protein sequence alignment. It is a putative sAgs. The authors didn’t present any experimental evidences to support the newly identified SpeS is a sAgs.
A major revision is required.
Author Response
Reviewer 3
- In this manuscript, the authors report the identification of new bacterial superantigens (sAgs) of S. zooepidemicus, SpeS, and show that mutation of the putative T cell receptor (TCR)-binding motif (YAY to IAY) abrogated TCR-binding, whilst maintaining interaction with major histocompatibility complex (MHC) class II molecules. It is interesting, but the authors missed a very important functional assay for the newly identified sAgs. SpeS was identified based on protein sequence alignment. It is a putative sAgs. The authors didn’t present any experimental evidences to support the newly identified SpeS is a sAgs. A major revision is required.
> In our manuscript we indeed report the in silico identification of SpeS, but we go on to demonstrate that recombinant SpeS induced a dose-dependent mitogenic response in equine PBMCs. We show that SpeS induced the production of IFNγ and bound to MHC Class II receptors. As this reviewer states, we also demonstrate that mutation of the putative T cell receptor binding motif abrogated the activity of SpeS in agreement with the mechanism of action of other known superantigens. Therefore, we (and the other two reviewers) believe that our manuscript demonstrates that SpeS is indeed a new superantigen and we respectfully disagree with the suggestion by this reviewer that we have not presented any experimental evidence to support this claim.
Round 2
Reviewer 3 Report
- This manuscript has been improved, and should be ready for publicing.
Author Response
- This manuscript has been improved, and should be ready for publicing.
> We thank the reviewer for their comments and are glad that our manuscript is now acceptable for publication.